# Comparative Analysis of *Actaea* Chloroplast Genomes and Molecular Marker Development for the Identification of Authentic Cimicifugae Rhizoma

**DOI:** 10.3390/plants9020157

**Published:** 2020-01-27

**Authors:** Inkyu Park, Jun-Ho Song, Sungyu Yang, Byeong Cheol Moon

**Affiliations:** Herbal Medicine Resources Research Center, Korea Institute of Oriental Medicine, Naju 58245, Korea; pik6885@kiom.re.kr (I.P.); songjh@kiom.re.kr (J.-H.S.); sgyang81@kiom.re.kr (S.Y.)

**Keywords:** *Actaea*, *Cimicifuga*, plastid, indel marker, herbal medicine

## Abstract

*Actaea* (Ranunculaceae; syn. *Cimicifuga*) is a controversial and complex genus. Dried rhizomes of *Actaea* species are used as Korean traditional herbal medicine. Although *Actaea* species are valuable, given their taxonomic classification and medicinal properties, sequence information of *Actaea* species is limited. In this study, we determined the complete chloroplast (cp) genome sequences of three *Actaea* species, including *A. simplex*, *A. dahurica,* and *A. biternata*. The cp genomes of these species varied in length from 159,523 to 159,789 bp and contained 112 unique functional genes, including 78 protein-coding genes, 30 transfer RNA genes, and 4 ribosomal RNA genes. Gene order, orientation, and content were well conserved in the three cp genomes. Comparative sequence analysis revealed the presence of hotspots, including *ndhC-trnV-UAC*, in *Actaea* cp genomes. High-resolution phylogenetic relationships were established among *Actaea* species based on cp genome sequences. *Actaea* species were clustered into each *Actaea* section, consistent with the Angiosperm Phylogeny Group (APG) IV system of classification. We also developed a novel indel marker, based on copy number variation of tandem repeats, to facilitate the authentication of the herbal medicine Cimicifugae Rhizoma. The availability *Actaea* cp genomes will provide abundant information for the taxonomic and phylogenetic analyses of *Actaea* species, and the *Actaea* (ACT) indel marker will be useful for the authentication of the herbal medicine.

## 1. Introduction

Chloroplasts play an essential role in photosynthesis and carbon fixation [1]. In angiosperms, chloroplast (cp) genomes range in size from 115–180 kb and exhibit a quadripartite structure consisting of two single copy, a large single copy (LSC) region, a small single copy (SSC) region, and two copies of an inverted (IR) region (IRa and IRb) [2,3]. In general, cp genomes of angiosperms contain 110–130 genes, with up to 80 protein-coding genes, 30 transfer RNA (tRNA) genes, and 4 ribosomal RNA (rRNA) genes [1]. The structure of the cp genome is highly conserved among plant species compared with that of nuclear and mitochondrial genomes [4]. Moreover, cp genomes exhibit maternal inheritance, thus facilitating the conservation of gene contents and genome structure [5,6]. Consequently, the cp genome is widely used for species identification, molecular marker development, evolutionary analysis and high-resolution phylogenetic analysis [7,8,9]. With advances in next-generation sequencing, the cost of cp genome assembly has decreased, while the speed of sequencing has increased [8]. Moreover, cp genome sequencing represents a viable alternative to DNA barcoding, a controversial method of species identification. Recently, complete cp genomes have been reported for four *Actaea* species, including *A. heracleifolia*, *A. dahurica*, *A. vaginata*, and *A. asiatica* [10,11]. Genomic information available for *Actaea* species identification is limited, and additional genomic information is needed to understand the utility of *Actaea* phylogeny and the evolutionary relationship among the *Actaea* species.

The genus *Actaea* L. (including *Cimicifuga* L. ex Wernisch. and *Souliea* Franch.) comprises approximately 27 species, which show a wide distribution in Asia, Eurasia, and North America [12]. Plants of *Actaea* species are perennial herbs, with racemose or paniculate inflorescences that bear many actinomorphic flowers with free carpels and follicular or baccate (berry-like) fruits [13]. However, the classification of *Actaea* and its related genera remains controversial. Compton et al. merged *Actaea* and *Cimicifuga* with *Souliea* based on sequence information of the internal transcribed spacer (ITS) region and the *trnL-F* gene [12]. Moreover, Compton et al. [12] identified seven sections within the genus *Actaea*. However, Wang et al. [14] proposed that *Actaea*, *Cimicifuga*, and *Souliea* were independent genera, based on the morphological, palynological, and cytological characteristics of these plant species. *Actaea* and *Cimicifuga* are currently recognized as members of the tribe Cimicifugeae [15,16]. Phylogenetic analysis based on the nuclear ITS region and cp loci revealed that *Actaea* and *Cimicifuga* form a monophyletic group [12,17,18]. The genetic relationship of *Actaea* and *Cimicifuga* with the closest genus *Eranthis* is highly similar. Thus, *Actaea* and *Cimicifuga* are together recognized as the genus *Actaea*, which was formed by merging *Cimicifuga* with *Souliea* [12].

Dried rhizomes of *A. simplex, A. dahurica,* and *A. heracleifolia* are used as a Korean traditional herbal medicine, namely, Cimicifugae Rhizoma, which is recognized as a medicine in the Korean Herbal Pharmacopoeia [19]. Cimicifugae Rhizoma is controlled by the Ministry of Food and Drug Safety due to its pharmaceutical properties [20,21,22]. Only the rhizomes of *A. simplex, A. dahurica,* and *A. heracleifolia* are considered as authentic Cimicifugae Rhizoma in Korea [19]. However, rhizomes of other *Actaea* species such as *A. biternata* and *A. asiatica*, which are also distributed in Korea [12,19], are morphologically highly similar to those of *A. simplex, A. dahurica,* and *A. heracleifolia* to the naked eye. Consequently, Cimicifugae Rhizoma has been adulterated with closely related other *Actaea* species in Korean herbal markets [23]. Thus, authentication of *Actaea* species is important for preserving a uniform pharmacological effect of Cimicifugae Rhizoma.

In this study, we determined and characterized the cp genomes of three follicular *Actaea* species, including *A. simplex*, *A. dahurica,* and *A. biternata*, and compared these sequences with the cp genome sequences of three additional *Actaea* species. Phylogenetic analysis was performed to establish the genetic relationships of *Actaea* species within Cimicifugeae. Moreover, we developed a novel indel marker to authenticate Cimicifugae Rhizoma as a herbal medicine. Our results contribute available data for the authentication of Cimicifugae Rhizoma, based on genomic features, which will help preserve the quality of this herbal medicine. Moreover, our results provide key insights into the evolution of species within Cimicifugeae.

## 2. Results and Discussion

### 2.1. Features of the cp Genomes of Three Actaea Species

Sequencing of the cp genomes of three *Actaea* species including *A. simplex*, *A. dahurica*, and *A. biternata* at approximately 31×, 74×, and 280× coverage, respectively, generated 2.1–2.3 Gb raw paired-end read data, which was equivalent to 1.4–1.7 Gb trimmed reads (Appendix A). The cp genomes of all three *Actaea* species were of high quality (Figure 1, Table 1). The complete cp genome sequences of the three *Actaea* species varied in length from 159,523–159,789 bp and showed a typical quadripartite structure, with the LSC region of 88,723, 88,652, and 88,958 bp, SSC region of 17,865, 17,729, and 17,757 bp and each IR region of 26,518, 26,571, and 26,537 bp in *A. simplex*, *A. dahurica*, and *A. biternata*, respectively. Junction regions in the cp genome of each species were validated by PCR-based sequencing (Appendix A, Appendix A). In addition, the sequence reads were validated by mapping them onto the complete cp genomes (Appendix A). The GC content of the cp genome of each species was approximately 38% (Table 1). In general, the GC content of IRs was higher than that of single copy regions in all three species [10,11]. Gene content and gene order were highly similar among the three *Actaea* species. The cp genome of each *Actaea* species harbored 112 unique genes, including 78 protein-coding genes, 4 rRNA genes, and 30 tRNA genes (Table 1, Appendix A). The cp genomes of *Actaea* species exhibited duplicate genes in the IR regions. In all three cp genomes, *rpl32* (large ribosomal protein 32) was a pseudogene, with partial deletion. This was consistent with a previous report of the loss or pseudogenization of *rpl32* within Ranunculaceae [11]. In the tribe Isopyreae, *rpl32* from the cp genome was integrated into the nuclear genome, demonstrating gene transfers from an organelle genome to the nuclear genome [11,24]. A total of 17 intron-containing genes were detected in the 3 cp genomes, including 11 protein-coding genes, 6 tRNA genes harboring a single intron and 2 genes (*ycf3* and *clpP*) harboring 2 introns (Appendix A). The cp genomes of all three *Actaea* species showed a stable and typical quadripartite structure similar to that of other angiosperms [25,26,27]. The genome length, gene content, gene orientation, and GC content of the three *Actaea* cp genomes were similar to those of genera within Cimicifugeae [11]. Analysis of codon usage and anticodon recognition patterns revealed that the three *Actaea* cp genomes contained 26,669–26,696 codons and codons for leucine (Leu), isoleucine (Ile), and serine (Ser) were the most abundant (Appendix A). The value of relative synonymous codon usage (RSCU) was greater than 5 for arginine (Arg), Leu and Ser, as expected (Appendix A). The RSCU values represented synonymous codon bias with a high proportion of A or T at the third position, similar to the cp genomes of other angiosperms [28,29,30]. The amino acid of high RSCU values infers that the contribution functionally prevents error in the peptide or transcription process [28,30]. Overall, the characterization of three *Actaea* cp genomes showed that these genomes are highly conserved, similar to the cp genomes of other *Actaea* species [10,11].

### 2.2. Repeat Sequences in cp Genomes of Three Actaea Species

SSRs are abundant in nuclear and organelle genomes and used in phylogenetic analysis and population genetics because of their stability and transferability [31,32]. A total of 168, 172, and 183 SSRs were detected in *A. simplex*, *A. dahurica*, and *A. biternata* cp genomes, respectively, using the Microsatellite Identification Tool (MISA) program [33] (Appendix A). The number of SSRs per unit length was higher in the single copy (SC) regions (LSC and SSC) than in IR regions in all three *Actaea* species. The cp genome of *A. biternata* contained numerous SSRs in IR regions compared with other *Actaea* species (Appendix A). The intergenic spacer (IGS) regions and exons in cp genomes contained mono- and dinucleotide SSRs. Tandem repeat sequences influenced the genomic structure variation and gene duplication and were often used for developing indel markers because of their copy number variation [4,27]. We surveyed tandem repeat sequences in cp genomes of *Actaea* species using the Tandem Repeats Finder program [34]. We detected 21, 23, and 20 tandem repeats in more 20 bp of *A. simplex* (Appendix A), *A. dahurica* (Appendix A), and *A. biternata* (Appendix A), respectively. These tandem repeats were 8–46 bp in length and were present as 2–4 copies in the three *Actaea* cp genomes (Appendix A). The majority of tandem repeats were located in the LSC region. We used these tandem repeats, particularly those in *rps16*-*trnQ-UUG*, for the development of an indel marker to identify authentic Cimicifugae Rhizoma.

### 2.3. Structure of cp Genomes of Actaea Species

To investigate divergence within the genus *Actaea*, we compared the structure of cp genomes of six *Actaea* species including *A. simplex*, *A. dahurica*, *A. biternata*, *A. heracleifolia*, *A. asiatica*, and *A. vaginata*; cp genomes of the latter three species, *A. heracleifolia* (NC_042253)*, A. asiatica* (NC_041525), and *A. vaginata* (NC_041543), were downloaded from GenBank database of the National Center for Biotechnology Information (NCBI). All six *Actaea* species showed greater divergence in the SC regions (LSC and SSC) than in the IR regions. The majority of divergent sequences were located in the IGS regions. In genic regions, *matK*, *ycf1*, and *ycf2* showed greater divergence than other genes; this is consistent with previous studies [35,36,37]. The *matK* gene harbors an important DNA barcode region for species identification [35]. All six *Actaea* species showed divergence in three regions including *trnC-GCA-petN*, *ndhC-trnV-UAC*, and *ycf1-ndhF*. Additionally, *A. biternata* and *A. asiatica* showed high divergence in the region from *petB* to *petD* (Figure 2). Divergence in the *ndhC-trnV-UAC* region, known as the hotspot region, has been detected in the cp genomes of other angiosperms [36,37]. These regions represent potential candidates for molecular marker development for species identification. We analyzed syntenic regions in *Actaea* species using the MAUVE algorithm. The cp genomes of all six *Actaea* species formed a well-conserved collinear block (Appendix A), thus showing a highly conserved structure.

To determine sequence divergence in the six *Actaea* cp genomes, we calculated the nucleotide diversity (*Pi*) (Figure 3). The total value of *Pi* for all six *Actaea* species at the cp genome level was 0.0104. The IR regions showed greater conservation than the SC regions, with average pi values of 0.0027 and 0.0234, respectively. The IGS region in *ndhC*-*trnV-UAC* showed high divergence, with a *Pi* of 0.0418. In the genic region, the value of *Pi* for *ndhF* was 0.01254, indicating that the genic region was more highly conserved than the IGS region, as expected. In this study, although the cp genomes of all six *Actaea* species showed a highly conserved structure, high sequence variability was detected at the genus level. The IR/SC border regions were highly conserved among the six *Actaea* species (Appendix A). The *rps19* genes of the three *Actaea* were located at the LSC/IRa region. The *ycf1* pseudogene and *ycf1* gene were located in the junction of IRa/SSC and SSC/IRb. The *trnH-GUG* genes were all located in the LSC region. Overall, cp genomes of all the six *Actaea* species showed a consistent structure. To identify genetic variation among the six species, we analyzed the non-synonymous to synonymous substitution ratio (Ka/Ks) using 67 cp genes conserved among *A. simplex*, *A. dahurica*, and *A. biternata* (Appendix A). Most of these genes were under purifying selection (Ka/Ks ≤ 0). The *accD* gene showed a Ka/Ks ratio of 2.156 in comparison with *A. dahurica* and *A. biternata*, while the *ycf2* gene showed a Ka/Ks ratio of 1.167 in comparison with *A. simplex* and *A. biternata*, implying that *accD* and *ycf2* were under positive selection, which led to greater environmental adaptation. The *accD* gene encodes acetyl-CoA carboxylase, a key enzyme that catalyzes the carboxylation of acetyl-CoA to produce malonyl-CoA [1]. Several species have lost the *accD* gene from their cp genomes, but the function of *accD* performed in the nuclear genome, performs the same function as *accD*. [7,38]. Positive selection of *accD* and *ycf2* genes has been reported in the cp genomes of other species within the genera *Ipomoea*, *Anemopaegma*, and *Amphilophium* [27,39,40]. Furthermore, *accD* and *ycf2* have been frequently reported as lost or highly variable [3].

### 2.4. Phylogenetic Relationship among Actaea Species

We determined the phylogenetic relationship of *A. simplex, A. biternate*, and *A. dahurica* with other *Actaea* species within the tribe Cimicifugeae based on two data sets: 65 conserved protein-coding gene sequences (Figure 4) and whole cp genome sequences (Appendix A). In both phylogenetic trees, *Actaea* species were clustered, consistent with the *Actaea* sectional classification of Compton et al. [12]. Five out of seven nodes in both phylogenetic trees showed maximum likelihood (ML) bootstrap (BS) values of 100% and Bayesian inference (BI) posterior probability (PP) values of 1.0. *A. heracleifolia* and *A. simplex* clustered within a monophyletic group, showed a sister relationship with *A. dahurica*. *A. vaginata* showed a paraphyletic relationship with the other five *Actaea* species, consistent with the APG IV system of classification [41]. Compton et al. [12] reclassified *Actaea* to include 27 *Cimicifuga* species, based on morphology, nuclear ITS region, and cp *trnL-F* region, and strongly supported the delimitation of *Actaea* as seven sections. Our phylogenetic analyses divided *Actaea* species into four groups, which was consistent with the previous infrageneric classification (*Actaea* sections *Actaea*, *Dichanthera*, *Pityrosperma*, and *Souliea*). Recently, phylogenetic analysis of 35 species within Ranunculaceae, based on cp genome sequences, showed that *A. dahurica* and *A. vaginata* formed a monophyletic cluster [11]. However, our results indicated that *A. dahurica* and *A. vaginata*, both of which are follicular species, were distinct, whereas *A. biternata* and *A. asiatica*, which produce different fruit types (follicle and berry, respectively), were the closest relatives.

To understand the phylogenetic relationship between *Actaea* and Cimicifuga, an in-depth investigation of other CP genomes and reinterpretation of morphological data are needed. We propose that further studies of *Actaea* with expanded species, including *A. japonica* from *Actaea* sect. *Pityrosperma* and *A. erythrocarpa* from *Actaea* sect. *Actaea* as well as the North American (sect. *Podocarpae* and *Oligocarpae*) and Eurasian *Actaea* (sect. *Cimicifugae*) species, be considered to evaluate the cp genome trends and their evolution. Overall, these results provide insights into the phylogenetic relationship among species within Cimicifugeae.

### 2.5. Development of an Indel Marker for the Identification of Authentic Cimicifugae Rhizoma

We detected sequence variations in the cp genomes of five *Actaea* species to distinguish between authentic and adulterated Cimicifugae Rhizoma. Results showed that cp genomes of *A. biternata* and *A. asiatica*, which are usually found as adulterants in Cimicifugae Rhizoma, carry 67- and 39-bp insertions, respectively, in the *rps16-trnQ-UUG* region, unlike *A. heracleifolia*, *A. simplex*, and *A. dahurica*, sources of authentic Cimicifugae Rhizoma. Indel primers were designed flanking *rps16* and *trnQ-UUG* genes (Table 2). The ACT primers successfully amplified cpDNA from five *Actaea* species (Figure 5). Twenty accessions of five *Actaea* were used and successfully distinguished between authentic sources of Cimicifugae Rhizoma (*A. heracleifolia*, *A. simplex* and *A. dahurica*) and adulterants (*A. biternata* and *A. asiatica*) (Figure 5 and Appendix A). We also detected copy number variations of tandem repeats in cp genomes of *Actaea*. While *A. heracleifolia*, *A. simplex*, and *A. dahurica* contained no tandem repeats, *A. biternata* and *A. asiatica* contained two and three copies of tandem repeats, respectively, indicating species-specific variation. Unfortunately, we did not detect copy number variation of tandem repeats within *Actaea* sect. *Dichanthera*, which was not found any divergence sequence for distinguishing among three *Actaea* species, *A. heracleifolia*, *A. simplex* and *A. dahurica* at PCR-based approach resulted in using 12 accessions. Samples used in this study were obtained from diverse native regions in Korea. To develop species-specific markers, a larger collection of samples is needed.

Indel variations have played an important role in plant genome evolution and are responsible for genomic rearrangements via slipped strand mispairing and stem-loop secondary structures via intramolecular recombination [42,43,44,45]. Previously, seeds of *Ipomoea nil* and *I. purpurea*, components of the herbal medicine Pharbitidis Semen, have been successfully distinguished from those of other related *Ipomoea* species using indel markers based on variation in cp genomes [27]. Indel markers have also been used for species identification in the genera *Chenopodium* (to distinguish between *C. quinoa* and *C. album*) [46] and *Fagopyrum* (to distinguish between *F. tatricum* and *F. esculentum*) [47]. Thus, indel markers have been successfully used for species identification and herbal medicine authentication, thus overcoming the limitation of conventional methods such as DNA barcoding. The ACT marker could be useful for the authentication of *A. heracleifolia, A. simplex,* and *A. dahurica* and authentication of Cimicifugae Rhizoma.

## 3. Materials and Methods

### 3.1. Plant Material

Plant material used in this study was collected from natural populations in Korea. Representative fresh leaves of *A. simplex* (37°09′00.1″ N and 128°54′14.3″ E), *A. dahurica* (37°07′10.0″ N and 128°56′24.9″ E), and *A. biternata* (33°21′59.1″ N and 126°26′21.0″ E) were used for cp genome sequencing. These samples were assigned identification numbers, and voucher specimens were deposited in the Korean Herbarium of Standard Herbal Resources (Index Herbarium code KIOM) at the Korea Institute of Oriental Medicine (KIOM, Naju, Korea). Plant samples used for cp genome sequencing and indel marker validation are listed in Appendix A.

### 3.2. Genome Sequencing and Assembly

DNA was extracted from leaf samples using DNeasy Plant Maxi Kit (Qiagen, Valencia, CA, USA), according to the manufacturer’s instructions. Illumina short-insert paired-end sequencing libraries were constructed and sequenced using the MiSeq platform (Illumina, San Diego, CA, USA). De novo and reference-guided strategies were used to assemble cp genomes. High quality paired-end reads were assembled using the CLC genome assembler (ver. 4.06 beta; CLC Inc., Rarhus, Denmark) with default parameters. Highly similar contigs representing cp genome sequences were retrieved using Nucmer [48], and mapped to the reference cp genomes of *Megaleranthis saniculifolia* (FJ597983) and *A. asiatica* (NC_041525). We determined the order of aligned contigs according to the reference cp genome, and gaps were filled using Short Oligonucleotide Analysis Package (SOAP) de novo gap closer, based on the raw reads to the assembly [49]. Four cp junctions (LSC/IRa, IRa/SSC, SSC/IRb, and IRb/LSC) were confirmed by PCR-based sequencing using sequence-specific primers to validate correct genome assembly (Appendix A). Finally, raw reads were mapped onto the complete cp genome sequences using BWA ver. 0.7.25 [50].

### 3.3. Genome Annotation and Comparative Analysis

The DOGMA and GeSeq [51,52] tools were used to annotate the three complete *Actaea, A. simples*, *A. dahurica,* and *A. biternata,* cp genomes. Start and stop codons of protein-coding sequences were manually corrected based on reference genomes (FJ597983 and NC_041525). The tRNAscan-SE 1.21 [53] was used to obtain and identify tRNA genes. The circular gene maps were constructed using the OrganellarGenomeDRAW (OGDRAW) [54]. MEGA6 was analyzed to GC content and codon usage of cp genomes [55]. MAUVE V2.3.1 was used to identify local collinear blocks of 6 *Actaea* species (*A*. *heracleifolia*, *A. simplex, A. dahurica*, *A. biternata, A. asiatica*, and *A. vaginata*) [56]. Nucleotide diversity (Pi) among the 6 *Actaea* species was calculated using DnaSP version 6.1 [57]. Pi value of 0 was excluded. Values of Ka, Ks and Ka/Ks ratio were estimated using the KaKs Calculator ver. 2.0 [58].

### 3.4. Repeat Analysis

The MISA tool was used to detect SSRs in cp genomes of *A. simplex*, *A. dahurica*, and *A. biternata* [33], with the minimum number of repeats set at 10, 5, 4, 3, 3, and 3 for mono-, di-, tri-, tetra-, penta-, and hexanucleotide SSRs, respectively. Tandem repeat finder was used to identify tandem and palindromic repeats (≥ 20 bp) with the following parameters: Minimum alignment score = 50; maximum period size = 500; repeat identity ≥ 90% [34].

### 3.5. Phylogenetic Analysis

In this study, 2 matrices were used for phylogenetic analysis: 1) A total of 65 conserved protein-coding sequences in 9 cp genomes aligned over a length of 66,374 bp; and 2) whole cp genomes. Six cp genome sequences within the tribe Cimicifugeae were downloaded from NCBI (Appendix A). *Beesia calthifolia* and *Anemonopsis macrophylla* were used as outgroups. Using MAFFT, 65 protein-coding genes and whole cp genomes were aligned [59], the gaps in the alignment were stripped using BioEdit [60]. The best-fitting model was selected based on the Akaike Information Content (AIC) using JModeltest V2.1.10 [61]. The GTR + I + G model was applied to protein-coding genes (Appendix A), and the TVM + I + G model was used for whole cp genomes (Appendix A). ML analysis was performed using MEGA6 [55], and branch support was calculated with 1000 bootstrap replicates. BI analysis was performed in MrBayes 3.2.2 [62] using the settings as following: Two independent Markov Chain Monte Carlo (MCMC) runs were performed for 1 million generations with samples every 1000 generations, the first 25% of trees were discarded as burn-in.

### 3.6. Development and Validation of the ACT Indel Marker

The indel region was detected based on mVISTA similarities and aligned sequences. To amplify these regions, primers were designed using Primer-BLAST (NCBI). ACT indel marker was amplified from 20 ng of genomic DNA in a 20-µL reaction volume containing Solg™ 2X *Taq* PCR Smart Mix 1 (Solgent, Daejeon, Korea) and 10 pmol of each primer (BIONICS, Seoul, Korea). Amplification of ACT was conducted on a C1000 Touch^TM^ Thermal Cycler (Bio-Rad Laboratories, Inc., Hercules, CA, USA) with the following conditions: Initial denaturation at 95 °C for 2 min, followed by 35 cycles of denaturation at 95 °C for 40 s, annealing at 60 °C for 40 s, and extension at 72 °C for 50 s and lastly a final extension at 72 °C for 5 min. PCR products were separated on 2% agarose gels at 150 V for 40 min. All samples were obtained from the KIOM herbarium (Appendix A).

## 4. Conclusions

In this study, we determined and analyzed the cp genome sequences of *A. simplex*, *A. dahurica,* and *A. biternata*. The cp genomes of all three species were highly conserved about the genomic characterization such as gene content and orientation, and GC content, although local sequence variations were also detected. Additionally, we established phylogenetic relationships among *Actaea* species using whole cp genome sequences and protein-coding sequences. The results of this study help determine the complex taxonomical classification of *Actaea*. Furthermore, based on copy number variations of tandem repeats in cp genomes of *Actaea* species, we developed an indel marker for species identification and authentication of herbal medicines. These results could add to the authentic herbal medicinal material containing *A. simplex, A. dahurica,* and *A. heracleifolia* such as Cimicifugae Rhizoma. These cp genome data of *Actaea* provide useful information to identify medicinal materials and hypothesize systematic and evolutionary works.

## Figures and Tables

**Figure 1 plants-09-00157-f001:**
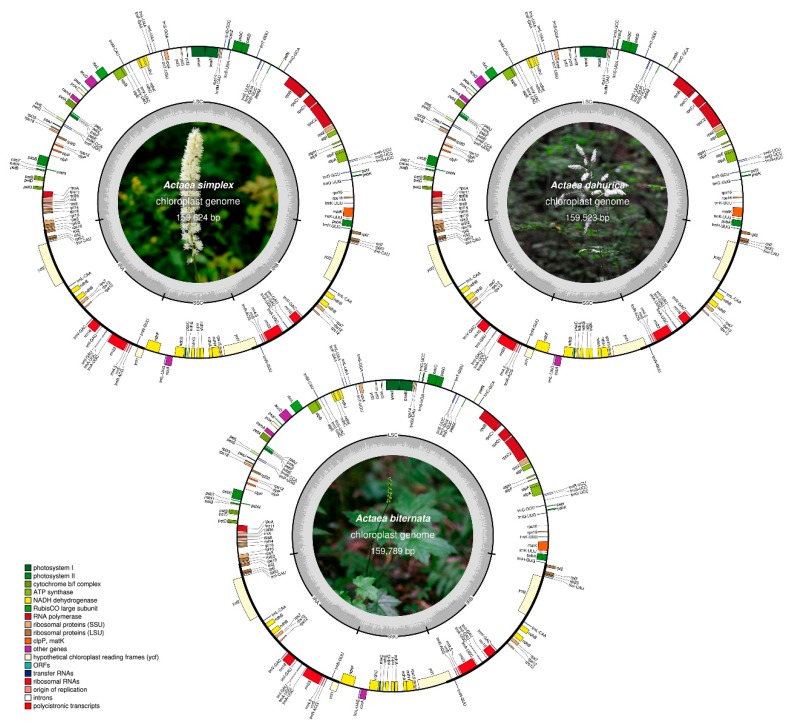
Graphical maps of the cp genomes of *A. simplex*, *A. dahurica,* and *A. biternata*. Genes drawn inside the outermost circle are transcribed clockwise, and those outside the circle are transcribed counterclockwise. Dark gray shading in the inner circle indicates the GC content. The inner photograph shows *Actaea*.

**Figure 2 plants-09-00157-f002:**
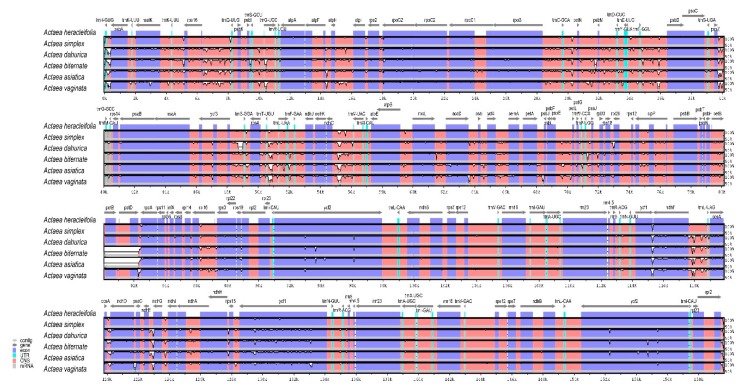
Comparative analysis of the cp genomes of six *Actaea* species, including *A. simplex*, *A. dahurica*, *A. biternata*, *A. heracleifolia*, *A. asiatica*, and *A. vaginata*, using mVISTA. The cp genome of *A. heracleifolia* used as a reference. Blue blocks, conserved genes; sky-blue blocks, tRNA and rRNA genes; red blocks, conserved non-coding sequences (CNSs); white blocks, regions polymorphic among the six *Actaea* species. A cut-off of 50% identity was used for the plots. The Y-axis represents a 50%–100% identity.

**Figure 3 plants-09-00157-f003:**
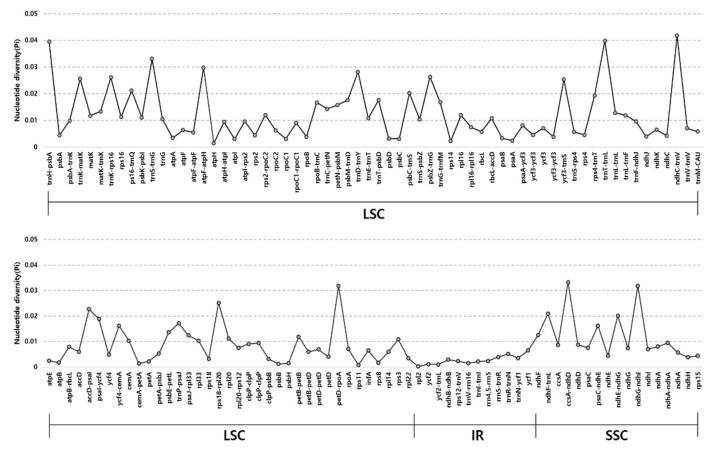
Comparison of nucleotide diversity (Pi) in the cp genomes of six *Actaea* species, including *A. simplex*, *A. dahurica*, *A. biternata*, *A. heracleifolia*, *A. asiatica*, and *A. vaginata*.

**Figure 4 plants-09-00157-f004:**
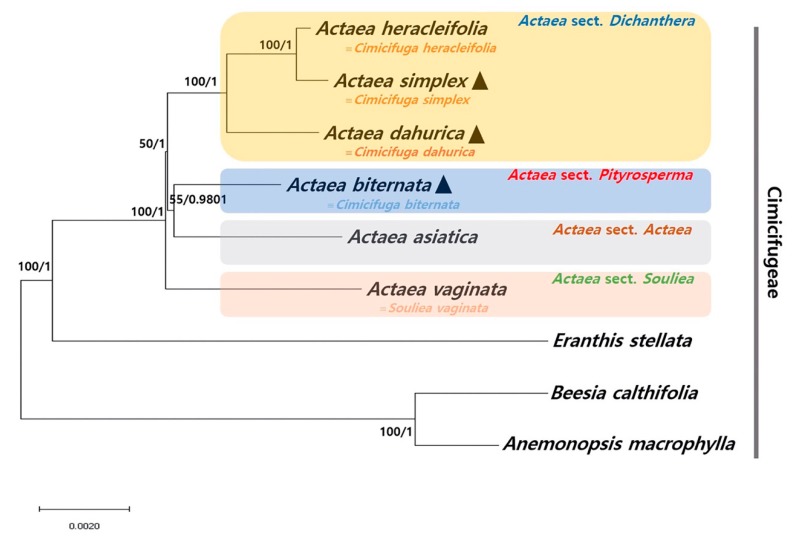
Phylogenetic analysis of nine species within the tribe Cimicifugeae. The phylogenetic tree was constructed with the maximum likelihood (ML) method and Bayesian Inference (BI) using 65 conserved protein-coding sequences from six *Actaea* species and one *Eranthis* species. *Beesia calthifolia* and *Anemonopsis macrophylla* were used as outgroups. ML topology is shown with bootstrap (BS) values (%), and BI posterior probability (PP) values at each node. Black triangles indicate the cp genomes of three *Actaea* species investigated in this study.

**Figure 5 plants-09-00157-f005:**
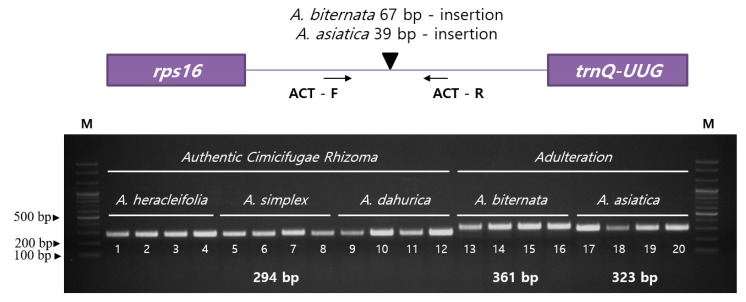
PCR amplification of the ACT indel marker in five *Actaea* species. The numbers indicate 20 accessions. Accession details are provided in Appendix A.

**Table 1 plants-09-00157-t001:** Characterization of the cp genomes of three *Actaea* species.

Characteristics-	*A. Simplex*	*A. Dahurica*	*A. Biternata*
GenBank accession number	MN623225	MN623226	MN623227
Chloroplast genome size (bp)	159,624	159,523	159,789
Large single copy (LSC) region (bp)	88,773	88,041	88,083
Inverted repeat (IR) region (bp)	26,518	26,571	26,537
Small single copy (SSC) region (bp)	17,865	17,729	17,757
Total number of genes	112	112	112
Number of protein-coding genes	78	78	78
Number of rRNA genes	4	4	4
Number of tRNA genes	30	30	30
GC content (%)	38.1	38.1	38.1
LSC (%)	36.2	36.3	36.2
IR (%)	43.1	43.1	43.1
SSC (%)	32.3	32.4	32.3

**Table 2 plants-09-00157-t002:** Primers used to develop the *Actaea* (ACT) indel marker.

Primer Name	Primer Sequence (5′→3′)	Position
ACT-F	TCA GCA TCG AGT TAG TAC CGT	*rps16-trnQ-UUG*
ACT-R	CCG AAT CGA GTA CCG ATG ACA	

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
