# Peer review of "Comparative Analysis of Actaea Chloroplast Genomes and Molecular Marker Development for the Identification of Authentic Cimicifugae Rhizoma"

_plants, 2020, doi:10.3390/plants9020157_

Round 1

Reviewer 1 Report

The authors of the article have determined and characterized the cp genomes of three follicular Actaea species. The three cp genomes were compared to additional Actaea species that are available in GenBank database. The authors performed a phylogenetic analysis to determine genetic relationships. Moreover, they have developed an indel marker in order to distinguish between authentic sources of Cimicifugae Rhizoma and adulterants. The results presented a valuable resource to identify Cimicifugae Rhizoma. However, I still have several concerns as indicated below.

Major comments

1- The cp genomes generated by the authors (A.simplex, A.dahurica, A.biternata) should be available in a database, like NCBI, and provide the codes in the paper. Table S10 provide a list of GenBank accession number but it can not be found A.simplex (MN623225), A.dahurica (MN623226), A.biternata (MN623227) in GenBank database, to what correspond these accession numbers?

2- The phylogenetic tree from Figure 4 should include one cp genome not from Cimicifugeae as an outgroup.

3- Line 229-230 and 237-238: It is not clear how many samples per accession were collected and from where to perform the PCR analysis. The authors should make it clear in the article and provide more information about samples and from where they have been collected. Are the samples from Table S9? Then, it could be useful to indicate in the PCR gel the number of each specimens from Table S9.

Minor comments

4- Line 142-143: The authors should indicate the number code of cp genomes from GenBank database.

5- Line 149: It is difficult to figure out which regions are petB and petD in Figure 2

6- Line 222-223: Why have the ACT indel marker not been amplified in A.vaginata? The authors should explain the reason for not including it.

7- Figure 1 and 2: The legend is too small

8- Figure 4: Indicate in the legend how many bootstraps were used in the phylogenetic tree

9- Figure 5: Instead of numbers, it could be more informative to indicate accession names. It is missing a blank control in the gel.

10- Figure S4: The authors should explain what represents white and green blocks.

11- Figure S7: It is missing the bootstrap value in the node which diverge Eranthis from Actaea

12- At the end of discussion: The authors should explain future perspectives and future projects.

Author Response

Review 1

The authors of the article have determined and characterized the cp genomes of three follicular Actaea species. The three cp genomes were compared to additional Actaea species that are available in GenBank database. The authors performed a phylogenetic analysis to determine genetic relationships. Moreover, they have developed an indel marker in order to distinguish between authentic sources of Cimicifugae Rhizoma and adulterants. The results presented a valuable resource to identify Cimicifugae Rhizoma. However, I still have several concerns as indicated below.

Major comments

1- The cp genomes generated by the authors (A.simplex, A.dahurica, A.biternata) should be available in a database, like NCBI, and provide the codes in the paper. Table S10 provide a list of GenBank accession number but it can not be found A.simplex (MN623225), A.dahurica (MN623226), A.biternata (MN623227) in GenBank database, to what correspond these accession numbers?

Response: We requested to NCBI and received release data schedule.

2- The phylogenetic tree from Figure 4 should include one cp genome not from Cimicifugeae as an outgroup.

Response: In this paper focused on relationships of Actaea species within Cimicifugeae. Our data is no problem. We sequenced three Actaea species and downloaded other three Actaea species from Genbank. Only six data had in NCBI. Thus, this study was used all of Cimicifugae cp genome for analyzing phylogenetic relationship Actaea species as well as Cimicifugae.

3- Line 229-230 and 237-238: It is not clear how many samples per accession were collected and from where to perform the PCR analysis. The authors should make it clear in the article and provide more information about samples and from where they have been collected. Are the samples from Table S9? Then, it could be useful to indicate in the PCR gel the number of each specimens from Table S9.

Response: Figure 5 revised and mentioned for Table S9. Revised manuscript Line 226

Minor comments

4- Line 142-143: The authors should indicate the number code of cp genomes from GenBank database.

Response: We show GenBank number for three Actaea species. Revised manuscript Line 139-141

5- Line 149: It is difficult to figure out which regions are petB and petD in Figure 2

Response: Figure 2 revised.

6- Line 222-223: Why have the ACT indel marker not been amplified in A.vaginata? The authors should explain the reason for not including it.

Response: Unfortunately, A. vaginata has not distributed in Korea. In Korea, Five Actaea, A. heracleifolia, A. simplex, A. dahurica, A. biternata, A. asiatica had. We mentioned authentic and adulterated Cimicifugae Rhizoma and 12 accessions in part of the Introduction and results and discussion. Revised manuscript Line 61-70, 227-232

7- Figure 1 and 2: The legend is too small

Response: We revised.

8- Figure 4: Indicate in the legend how many bootstraps were used in the phylogenetic tree

Response: We mentioned material and method. Revised manuscript Line 297-298

9- Figure 5: Instead of numbers, it could be more informative to indicate accession names. It is missing a blank control in the gel.

Response: We revised Figure 5. And this marker shows that is amplified all of 20 accessions. Do not need to blank control.

10- Figure S4: The authors should explain what represents white and green blocks.

Response: We revised Figure S4 legend.

11- Figure S7: It is missing the bootstrap value in the node which diverge Eranthis from Actaea

Response: We revised Figure S7

12- At the end of discussion: The authors should explain future perspectives and future projects.

Response: We mentioned for further study. Revised manuscript Line 202-208

Reviewer 2 Report

This study reported complete chloroplast (cp) genomes of three Actaea species: A. simplex, A. dahurica, and A. biternata. The authors clearly presented the features of cp genomes of these three species, performed comparative and phylogenetic analyses with other published Actaea species. English writing is good and results were well supported by tables, figures, and supplemental materials. In general, I enjoyed reading this manuscript and I think this manuscript is a good addition to the plant genomics resource. I only have a few suggestions regarding to grammatical errors and formatting:

Ln 57: These results showed... Please also double check other sentences.

Ln 97-99: "large ribosomal protein 32" should be written upon the first time "rpl32" was mentioned.

Author Response

Review 2

Comments and Suggestions for Authors

This study reported complete chloroplast (cp) genomes of three Actaea species: A. simplexA. dahurica, and A. biternata. The authors clearly presented the features of cp genomes of these three species, performed comparative and phylogenetic analyses with other published Actaea species. English writing is good and results were well supported by tables, figures, and supplemental materials. In general, I enjoyed reading this manuscript and I think this manuscript is a good addition to the plant genomics resource. I only have a few suggestions regarding to grammatical errors and formatting:

Response: Thank you for your positive comment.

Ln 57: These results showed... Please also double check other sentences.

Response: We changed other sentence. Revised manuscript Line 58

Ln 97-99: "large ribosomal protein 32" should be written upon the first time "rpl32" was mentioned.

Response: We revised. Revised manuscript Line 95

Round 2

Reviewer 1 Report

The manuscript has improved after first round of revisions. The authors have improved the text and have clarified about methods and samples used.

Besides that, I still have few comments but just regarding to grammatical errors and legend sizes.

1- Across the manuscript are used Cimicifugeae and Cimicifugae, please be consistent by writing in the same way

2- Line 324-325: check sentence

3- Figure 1 and 2: Legend is still too small

Author Response

Review 1

The manuscript has improved after first round of revisions. The authors have improved the text and have clarified about methods and samples used.

Besides that, I still have few comments but just regarding to grammatical errors and legend sizes.

1- Across the manuscript are used Cimicifugeae and Cimicifugae, please be consistent by writing in the same way

Response: Cimicifugeae, Cimicifugae, Cimicifugae Rhizoma is different sense. We confirmed as to  whether correct or not.

Cimicifugeae is indicated tribe in taxonomic level. Cimicifugae is section, intalic, in taxonomic level. Cimicifugae Rhizoma : herb medicine name

2- Line 324-325: check sentence

Response: We revised it.

3- Figure 1 and 2: Legend is still too small

Response: We revised Figure 1, and original figures files have attached. Please confirme that. Attached file is high resolution.
